# Clinical Applications for Gasotransmitters in the Cardiovascular System: Are We There Yet?

**DOI:** 10.3390/ijms241512480

**Published:** 2023-08-05

**Authors:** Elisa Arrigo, Stefano Comità, Pasquale Pagliaro, Claudia Penna, Daniele Mancardi

**Affiliations:** Department of Clinical and Biological Sciences, University of Torino, 10124 Turin, Italy; elisa.arrigo@unito.it (E.A.); stefano.comita@unito.it (S.C.); claudia.penna@unito.it (C.P.)

**Keywords:** hydrogen sulfide, nitric oxide, carbon monoxide, ischemia, reperfusion, cardioprotection, clinical trials

## Abstract

Ischemia is the underlying mechanism in a wide variety of acute and persistent pathologies. As such, understanding the fine intracellular events occurring during (and after) the restriction of blood supply is pivotal to improving the outcomes in clinical settings. Among others, gaseous signaling molecules constitutively produced by mammalian cells (gasotransmitters) have been shown to be of potential interest for clinical treatment of ischemia/reperfusion injury. Nitric oxide (NO and its sibling, HNO), hydrogen sulfide (H_2_S), and carbon monoxide (CO) have long been proven to be cytoprotective in basic science experiments, and they are now awaiting confirmation with clinical trials. The aim of this work is to review the literature and the clinical trials database to address the state of development of potential therapeutic applications for NO, H_2_S, and CO and the clinical scenarios where they are more promising.

## 1. Introduction

### 1.1. Gases as Signaling Molecules

Signaling molecules can be classified according to different criteria such as chemical properties, modalities of action, site of production, and targeting tissue, to name a few. Chemical complexity can span from simple ions—e.g., K^+^ is considered an endothelium-derived hyperpolarizing factor [1]—to highly articulated proteins [2]. Diffusion through lipids, water solubility, steric hindrance and charge, and structural stability account for signaling patterns that can be self-directed (autocrine), neighboring cell-oriented (paracrine), or blood-carried (endocrine). Over the last decades, a subfamily of signaling molecules has been clustered based on physicochemical and biological features, and they have been demonstrated to play a pivotal role in the cardiovascular system, both under physiological and pathological conditions [3]. This group of molecules shares chemical properties (gaseous form under normobaric and normothermic conditions), constitutive production, and a dual role in mediating basic functions as well as deleterious effects. The name *gasotransmitters* was proposed to indicate a family composed, for now, of three members: nitric oxide (NO), hydrogen sulfide (H_2_S), and carbon monoxide (CO). Despite their sometimes overlapping features [4], respiratory gases such as molecular oxygen and carbon dioxide are not included in the gasotransmitter trio. The role of gasotransmitters is, indeed, pleiotropic with reportedly significant actions in virtually all body compartments. Nevertheless, the majority of research, as well as the first descriptions, referred to the effects on the heart and blood vessels. Consistently, several scientists have focused their work on the development of new therapeutic strategies against cardiac ischemia/reperfusion (I/R) injury based on the growing knowledge about the role of gasotransmitters in myocardial infarction. The exact role, function, and interaction of gasotransmitters in the cardiovascular system are still under intense investigation using basic research. However, translational approaches and clinical applications have begun to provide substantial feedback that contributes to redirecting basic questions about their fine signaling mechanisms. One critical aspect of gasotransmitters is their double-edged effect of high deleterious levels and fundamental mechanisms of low basal concentrations. This double-faced feature is involved in both cytoprotective and cytotoxic outcomes in cardiomyocytes during ischemia and reperfusion, accounting, therefore, for cell fate during infarction. These aspects profoundly affect the design of clinical approaches in order to fully exploit the therapeutic potential of gasotransmitter-related drugs.

### 1.2. Gasotransmitter ID

In order to be considered a member of the gasotransmitter family, a molecule must be endowed with specific features, including:Being gaseous at temperature and pressure compatible with life;Dimensions and physicochemical features (low molar mass, between 28 and 34 g/mol; low solubility in water, between 0.004 and 3.980 g/L) allowing permeability through cellular outer and inner membranes (no receptor required for action);Regulated endogenous production by constitutively expressed enzymes (Figure 1);Basal levels exert fundamental and well-defined physiological functions;Toxic effects at high concentrations;Targeting of different organs/systems;Interaction with one or more member(s) of the family;Exclusion criteria: being a respiratory gas.

After deepening the investigation into the biological actions of hydrogen sulfide, a short list of requirements was proposed to allow new membership in the family [3]. Since this early stage, the requisites to be considered a member of gasotransmitters have grown stricter as new mechanisms and biological effects have been described.

## 2. Hydrogen Sulfide

### 2.1. H_2_S Biology in the Heart and Vessels

H_2_S was the last member to join the gasotransmitter family. Long known because of its odoriferous and noxious properties at less than 1 ppm, H_2_S is poisonous when inhaled at high doses. The severity of the symptoms is concentration- and time-dependent. Specifically:Between 50 and 100 ppm: mild conjunctivitis and airway irritation after an hour. May cause loss of appetite;Between 100 and 150 ppm: loss of smell (“*olfactory fatigue*”);Between 150 and 300 ppm: severe conjunctivitis and airway irritation after one hour. If the exposure is prolonged, it may cause pulmonary edema;Between 500 and 700 ppm: collapse within 5 min, eye damage within 30 min, and death after 30–60 min;Between 700 and 1000 ppm: rapid unconsciousness, immediate collapse (“*knockdown*”) in one or two breaths, and death within a few minutes;Between 1000 and 2000 ppm: instant death [5,6].

The toxicity of H_2_S is mainly due to the inhibition of cytochrome c oxidase of the mitochondrial electron transport chain through the competitive and reversible binding of sulfur to the FeII-CuI active sites of the molecule [7]. The first evidence of its specific effects on vessels dates back to the late 1990s, while, in the early 2000s, its role as a relevant gaseous signaling molecule was definitely confirmed using several independent studies [8]. By that time, it became clear that H_2_S is not just a by-product of cysteine metabolism (Figure 1), but it is of physiological relevance in several biological processes. The malodorous molecule is pivotal to maintaining normal arterial blood pressure, as described in mice lacking cystathionine-γ-lyase (CSE), which is a constitutively expressed H_2_S-producing enzyme [9]. This fundamental physiological effect is mediated by the activation of several membrane channels, although opposite modulation of vascular tone has been reported depending on the arterial beds and the experimental model [10]. Data about arterial vasoconstriction are scarce and lack solidity on mechanisms, including only a few studies on rat cerebral arteries [11], dual modulation of H_2_S on rat-isolated gastric arteries [12], and an oxygen-related vasomotor role [13]. Most of the evidence points to a ubiquitous vasorelaxant action of both endogenous and exogenous (micromolar range) H_2_S with the mediation of smooth muscle potassium channels leading to hyperpolarization/relaxation [14]. Indeed, several ion channels are involved in the vascular signaling of H_2_S including ATP-sensitive K^+^ (K_ATP_) channels [14], voltage-gated K^+^ channels [15], and 4-aminopyridine-sensitive voltage-gated potassium channels [16]. In 2014, Eberhardt et al. suggested that two gasotransmitters, NO and H_2_S, react to produce nitroxyl (HNO). They discovered that the transient receptor potential channel A1 (TRPA1) colocalizes with the generation of H_2_S and NO and that HNO activates TRPA1 by forming amino-terminal disulfide bonds, which causes a persistent calcium influx in the sensory chemoreceptor. Calcitonin gene-related peptide (CGRP) is consequently released, causing both local and systemic vasodilation. The generation of NO and the activation of the HNO-TRPA1-CGRP pathway are key factors in the vasorelaxant effects of H_2_S. According to their hypothesis, the neuroendocrine HNO-TRPA1-CGRP signaling pathway is a crucial component in the regulation of vascular tone throughout the cardiovascular system [17]. While the specific mechanism underlying each channel modulation is still controversial, some authors hypothesize that the regulation of potassium currents could be dependent on the S-sulfhydration of two specific extracellular cysteine residues in the channel’s sulfonylurea receptor (SUR) subunits [18,19]. Although potassium currents seem to mediate the vasodilator effect of H_2_S independently from calcium signaling [20], some results reported the involvement of large-conductance calcium-dependent potassium (K_Ca_) channels [21,22], suggesting an interplay between potassium and calcium homeostasis [8]. It is well established that H_2_S acts through the regulation of calcium channels in several cell types, both under physiological [23] and pathological conditions [24], and its role within the cardiovascular system is also strictly related to its effects on endothelial cells [25]. In the endothelium, H_2_S can react with NO through different mechanisms (see Section 5), leading to a complex signaling cascade that, in turn, can regulate smooth muscle contraction/relaxation [26]. The intricate signaling network of gasotransmitters in the cardiovascular system is thus dependent on the pleiotropic action(s) on different cells (endothelial, smooth muscle, and myocardial cells), different endogenous production, and the co-existence of other factors (endothelial dysfunction and other co-morbidities) leading to several potential therapeutic and pharmacological targets [27].

### 2.2. H_2_S and Ischemia/Reperfusion Injury

As a regulator of vascular tone and calcium and potassium homeostasis, H_2_S has been suggested as an important player in I/R injury. As such, the role of H_2_S in the cardiomyocytes hypoxia/reoxygenation *scenario* has been deeply investigated, leading to the description of cytoprotective antioxidant properties mediated by Ca^2+^ channels [28], the activation of the JAK2/STAT3 signaling pathway (pivotal element of the survivor activating factor enhancement (SAFE) cascade) [29], and inhibition of cardiomyocyte apoptosis and autophagy [30]. Consistently, the beneficial impact of H_2_S in mitigating ischemic consequences has been proven effective in terms of reduced myocardial infarction, decreased arrhythmia onset, impaired development of cardiac fibrosis, and hypertrophy reduction with a marked decrease in heart failure prognosis [31]. A negative correlation between H_2_S plasma levels and infarct size and mortality has been reported in rats, where left ventricle infarct size and mortality were significantly increased after inhibition of endogenous production through the inhibition of CSE with propargylglycine [32]. These results are in line with previous studies reporting the role of the CSE/H_2_S pathway in mediating the pathophysiological effect of isoproterenol-induced myocardial injury. Consistently, the same study provided an early demonstration of the cytoprotective effect of exogenous H_2_S administration, leading to a reduction in lipid peroxidation [33].

The description of the role of exogenous H_2_S in protecting the heart against I/R injury has drawn attention to releasing derivatives. In order to enhance the pharmacological efficacy of H_2_S-releasing moieties beyond a local vasodilator mediator, the subtle mechanisms underlying its cytoprotective role must be investigated further to provide insights on intracellular targets and describe the molecular pathways preferentially targeted by post-transcriptional effects such as S-sulfhydration [18,34]. By producing H_2_S, which reduces oxidative damage, mitochondria-targeted H_2_S donors are hypothesized to defend against acute I/R injury. Nevertheless, Miljkovic and colleagues created a mitochondria-targeted agent called MitoPerSulf that very quickly releases H_2_S within mitochondria to address the problem regarding the slow rate of H_2_S release by current donors after reperfusion. Rapid mitochondrial uptake of MitoPerSulf causes it to interact with endogenous thiols to produce a persulfide intermediate that releases H_2_S. MitoPerSulf protects against cardiac I/R injury in mice with the acute formation of H_2_S that slows respiration at cytochrome c oxidase and prevents mitochondrial superoxide production by reducing the membrane potential. A novel class of therapeutic medicines for the immediate management of I/R injury includes mitochondria-targeted substances that rapidly produce H_2_S [35]. Other concerns must be considered when using a translational approach, such as the actual basal level of the gasotransmitter. Many articles have mentioned abundant basal levels of H_2_S, often reporting a 50–150 μM range [36,37,38], whereas other authors have questioned this statement based on the fact that plasma samples are generally odorless without the characteristic unpleasant H_2_S smell that is common in physiological buffers with similar concentrations [39]. Filipovic et al. reviewed the enzymatic pathways for the synthesis and oxidation of H_2_S as well as its physiologically relevant chemistry. They explored the roles attributed to protein persulfidation in cell signaling pathways before discussing the chemical biology of persulfides and the chemical probes for detecting them [40]. Most recently, researchers have suggested that the reference level for baseline production of endogenous H_2_S can be set in the nanomolar range [41]. Addressing a reliable baseline level of sulfide in cells, tissues, and blood samples is of pivotal importance in developing therapeutic strategies relying on the manipulation of both endogenous and exogenous H_2_S and, thus, requires additional investigation. Post-translational modifications (PTMs) have the potential to serve as disease biomarkers and provide clinical insights, according to recent evaluations in cardiovascular disease. Although the development of computational techniques that combine protein interaction data with PTM abundance ratios to provide functional predictions represents a significant advancement, function prediction is still a challenging process. At many different levels, including the performance of the prediction tools, expected site assignments, and predicted enzyme activity, validation of the results will still be crucial [42].

### 2.3. H_2_S and Mitochondria

The main toxicological effect of high concentrations of H_2_S, which has been known for centuries, takes place in the mitochondria (Figure 2), where it acts at the level of the respiratory chain, through reversible inhibition of cytochrome c oxidase [5]. In the study proposed by Vitvitsky et al., it was shown that the reduction of ferric cyt c by H_2_S exhibits hysteretic behavior. This suggests an involvement of reactive sulfur species in the reduction mechanism and correlates with a reaction stoichiometry of 1.5 mol of cyt c reduced/mol of H_2_S oxidized. Human cell models (HT29 and HepG2) treated with the complex III inhibitor antimycin A experience an increase in O_2_ consumption in the presence of H_2_S, a condition compatible with the entry of sulfide-derived electrons at the level of complex IV. Protein persulfidation was induced in vitro by cyt c-dependent H_2_S oxidation. Conversely, when cyt c expression was silenced, protein persulfidation was reduced [43]. Domán and colleagues reviewed the mechanistic features, sulfide-mediated signaling, and the pathological effects of the interactions between reactive sulfur species (RSS) and protein metal centers. They concluded that the creation of H_2_S-related drugs and the development of redox medicine require a greater understanding of the molecular basis underlying the biological effects of reactive sulfur species [44]. Mitochondria are among the primary targets/mediators in the H_2_S signaling cascade, as demonstrated by their impact on apoptosis, biogenesis, dynamics, and morphology, in addition to the inhibition of the respiratory chain. Most interestingly, the interplay between mitochondria and H_2_S is complex and spans from vital bio-molecular functions to deleterious interference according to the level of exposure. In fact, the cytoprotective signaling cascade might be initiated by transient inhibition(s) of the respiratory chain, which can be reversed by the sulfide oxidation pathway, leading to metabolic reprogramming, as demonstrated in colonocytes [45]. Consistently, the biological response to H_2_S often shows a bimodal activity, with high levels causing inflammation [46], coronary artery diseases [47], septic shock [48], increased migration of tumor-derived endothelial cells [24], supported growth and proliferation of colon cancer [49], and low levels leading to elevated arterial blood pressure [50], accelerated progression of atherosclerosis [51], and facilitated pulmonary fibrosis disease [52]. A common player within this broad spectrum of biological effects is the mitochondrion, where protein persulfidation is initiated by the activity of sulfide quinone oxidoreductase (SQOR) [53]. SQOR is bonded to the inner membrane of mitochondria and acts as a regulator of H_2_S signaling, discriminating between physiological and poisoning outcomes [45]. SQOR can distinguish between the normal effects of H_2_S signaling (5–50 µM) and toxicity (over 200 µM). It accomplishes this by controlling the body’s production of H_2_S. SQOR lowers H_2_S synthesis to minimize toxicity when H_2_S concentrations are too high and promotes H_2_S generation to carry out its physiological tasks when levels are within the normal range [54]. Specifically, the proposed regulatory mechanism is structured as follows. First, SQOR fixes H_2_S to sulfite, resulting in the production of thiosulfate. Then, thiosulfate sulfur transferase (TST) mediates the production of a persulfide by the transfer of sulfur from thiosulfate to an SH-containing molecule (e.g., GSH). A persulfide dioxygenase (PDO), by extracting and oxidizing the sulfur atom from persulfide, leads again to the production of sulfite, which is further oxidized into sulfate by the enzyme sulfite oxidase (SOx) [55,56]. Libiad and coworkers also discovered that rhodanese primarily synthesizes thiosulfate instead of using it and that glutathione also serves as a persulfide acceptor for human SQOR [57]. Intriguingly, H_2_S is also produced in the mitochondria (Figure 1) by one of the two 3-mercaptopyruvate sulfur transferase (MPST) isoforms, MPST2, with a pattern of differential activity in line with the one reported for cystathionine-β-synthase (CBS) and CSE [58] and showing a significantly higher persulfide production in the liver, followed by the kidney and brain [59]. In humans, MPST expression has been detected in the kidney, liver, heart, and neurons, with its predominant expression in the gastrointestinal tract [60]. Some authors suggest that MPST can account for a prevalent H_2_S biosynthesis in tissues with a high content of cysteine, where its role in sulfur amino acid catabolism is pivotal [61]. In addition, MPST seems to be predominantly modulated through redox-sensitive processes because its oxidant-driven homodimer is assembled by disulfide bonds depending on the redox state [62]. When the production/consumption balance maintains H_2_S levels below 20 μM, the gasotransmitter can act as an inorganic substrate, granting electrons to the respiratory chain and potentially aiding oxidative phosphorylation and ATP production [63]. Indeed, H_2_S can participate in sustaining metabolism, and the portion generated by MPST2 has been demonstrated to be adequate to support electron flow in the respiratory chain and cellular bioenergetics [64]. In support of these data, mitochondrion-specific H_2_S releasers have been developed and shown to stimulate endothelial cell bioenergetics, with cytoprotective effects against the loss of mitochondrial DNA integrity [65]. MitoPerSulf, a mitochondria-targeted drug developed by Miljkovic and colleagues, as previously mentioned, allowed them to quickly target H_2_S in the mitochondria while masking a reactive persulfide moiety. Their most important discovery was that MitoPerSulf was initially protective in the in vivo left anterior descending artery (LAD) model of cardiac IR injury [35].

### 2.4. H_2_S and Clinical Applications

The clinical applications of H_2_S-based treatment are supported by the large body of evidence from basic research and indirect measurements of sulfur metabolism in patients affected by cardiovascular diseases (Table 1). It has been shown, in fact, that plasma levels of H_2_S are significantly lower in patients with coronary artery disease, coronary artery occlusion, and hypertension (Figure 3). Moreover, plasma H_2_S levels are inversely correlated with blood glucose while no significant change was associated with sex, age, cholesterol, triglyceride, low/high-density lipoprotein, or body mass index [66].

The described biological effects of H_2_S suggest that two putative strategies should be considered when developing sulfide-based therapeutic approaches: modulation of endogenous sulfide production through inhibition of CSE, CBS, and MPST and delivery of exogenous H_2_S with gas inhalation or specific sulfide donors. Therefore, it is crucial to keep in mind that CSE and CBS play significant roles in mammalian cysteine metabolism while seeking to optimize and further develop such compounds. In fact, homocysteinemia, a disease associated with high serum levels of homocysteine and homocystine (as well as related mixed disulfides), is known to result from genetic CBS deficiency and to cause endothelial dysfunction and hypertension [9]. To date, the production of H_2_S by 3MPST can be supported by the activity of D-amino acid oxidase (DAO), an enzyme mainly localized in mammalian peroxisomes of the central nervous system [67] and hepatocytes [68]. More recently, this pathway has been shown to significantly contribute to the production of H_2_S in the cerebellum and kidney of mice with an age-dependent pattern of DAO expression and activity. The same study confirmed the protective effects of endogenous H_2_S against oxidative stress in the observed tissue [69]. Although these findings support the pleiotropic importance of H_2_S-producing enzymes throughout mammalian organisms, there is no direct evidence for their pivotal role within the cardiovascular system, and further investigations are needed to determine the precise role of the pathway in the heart and vessels [70].

A cryoprotected phenotype, which is conferred by hibernation, is known to make tissues from hibernating animals extremely resistant to various hypoxic and ischemic insults [71]. Sulfide’s effects could also be studied in relation to the preservation of transplantable organs. In fact, hibernating animals’ organs are shielded from damage during storage and transplantation [72]. Heart failure patients commonly have lower plasma levels of H_2_S despite normal access to dietary sulfur. This deficit may be held responsible for increased oxidative stress exacerbating the pathophysiological processes contributing to heart failure. A clinical trial addressed the potential for supplementation with SG1002, a synthetic sulfur molecule positively tested in pre-clinical models [73], which can compensate for defective blood levels. Although this study primarily evaluated the safety and efficacy of multiple oral doses of SG1002 in subjects with heart failure, it also provided data about free H_2_S and its metabolites in treated patients when compared to the control/placebo group [74].

H_2_S donors have been conjugated with other pharmacological agents such as non-steroidal anti-inflammatory drugs (NSAIDs) [75]. In a recent clinical trial, it was reported that 4 and 24 h treatments with ATB-346, a H_2_S-releasing non-steroidal anti-inflammatory drug, displayed an increased blood flow compared with both untreated controls and those treated with an NSAID alone. The study was performed in a UV-KEc-triggered model of acute dermal inflammation, and the authors underlined the confusing observation about a counterintuitive increase in blood flow correlated with a significant reduction in reported pain and a lower tenderness score [76]. A previous study by the same group tested the ability of ATB-346 to lower gastrointestinal toxicity compared with naproxen, another equipotent COX inhibitor, and found encouraging results [77]. Although scarce, the results from these trials are promising and suggest that the use of exogenous donors of H_2_S to treat cardiovascular diseases is supported by preliminary clinical observations and growing evidence from basic science. Currently, larger, placebo-controlled, phase II and III studies are proposed for heart failure patients and also targeting comorbidities [76].

An alternative therapeutic approach based on sulfur supplementation has historically been performed using sulfurous thermal water exposure. Based on the observation that hydrotherapy can regulate leukocyte profiles and the proliferative response of T lymphocytes [78], it was demonstrated that sulfurous thermal water acts against inflammation by modulating the immune balance, which is triggered by an increase in anti-inflammatory cytokines and upregulation of antioxidant enzymes activity [79]. These results are in line with other pathological conditions, such as chronic obstructive pulmonary disease, where inhalation of thermal-derived reducing agents induced a significant reduction in ROS exhalation with a significant improvement in the respiratory distress assessment test [80].

## 3. Nitric Oxide (NO)

### 3.1. NO Biology in the Heart and Vessels

Three isoforms of the enzyme nitric oxide synthase (NOS) catalyze the tetrahydrobiopterin (BH4) and nicotinamide-adenine dinucleotide phosphate (NAD(P)H)-dependent oxidation of L-arginine to L-citrulline (Figure 1) and produce NO, a diatomic free radical gas, as one of the reaction products [81]. The three isoforms of NOS include two calcium-dependent isoforms, neuronal NOS (nNOS) and endothelial NOS (eNOS), and one calcium-independent isoform, inducible NOS (iNOS). The regulation of NOSs is particularly complex [82,83] and is summarized in Table 2. eNOS is mainly regulated by phosphorylation of its amino acid residues at multiple sites on its domains (e.g., T495, T497, S615, S633, Y657, and S1179), and the interaction between the enzyme and various other proteins and the different distribution in intracellular compartments adds a further degree of complication to this regulation. Factors that can influence NOS biology include calmodulin, caveolin-1, HSP90, CAT-1, CHIP (carboxyl terminus of HSP70-interacting protein), endoglin, GPCR (B2), GRX-1, β- actin, HDAC3 NOSIP, NOSTRIN, PIN1, PP1, PP2A, SIRT1, and VDAC1/2 [81,82,83,84,85,86,87,88,89,90]. For example, HSP90 plays an important role in NO synthesis since HSP90 silencing and its inhibition destabilize eNOS dimers, leading to degradation. Indeed, monomeric eNOS cannot deliver electrons to heme, but it transfers the electrons to a different location interacting with oxygen, thus facilitating the formation of superoxide (O_2_^−^). Consequently, eNOS dimerization is essential for proper enzymatic activity and NO production. In the absence of L-arginine or BH4, eNOS is uncoupled and the monomeric form of eNOS synthesizes O_2_^−^ in preference to NO [86]. Endothelin-1 can also facilitate O_2_^−^ production via eNOS in human microvascular endothelial cells when subjected to endotoxins [89]. Phosphorylation, however, does not appear to play a substantial role in this process since the state of phosphorylation has no effect on the dimer/monomer ratio of eNOS in endothelial cells or in cell-free preparations of purified eNOS protein [86]. Nevertheless, phosphorylation plays a pivotal role in determining the function of dimeric NOS. Indeed, phosphorylation of eNOS at Y657 and/or T495 leads to the downregulation of eNOS activity, whereas phosphorylation of S615/S633 increases eNOS calcium sensitivity and activity [85,87]. Several signals, including mechanical shear stress, bradykinin, adenosine, VEGF, statins, and 8-Br-cAMP, lead to the phosphorylation or de-phosphorylation of sites on eNOS by kinases/phosphatases. Each kinase has a specific site; for instance, T495 is phosphorylated by PKC, S615 by Akt, S633 by PKA and Pim1, and the Y657 site is phosphorylated by PYK2 [84,88]. NO can also derive from non-enzymatic processes [91].

The main intracellular target of NO is soluble guanylate cyclase (sGC), which, once activated, converts GTP to cyclic guanosine monophosphate (cGMP). This has several effects, including the inhibition of platelet aggregation and leukocyte adhesion and the relaxation of vascular smooth muscle, thus facilitating blood flow through different modalities. In arterioles and microvessels, NO is a powerful vasodilator, and its basal function consists of limiting the constriction of the vessels and remodeling the vessel walls. In capillaries, NO, in particular, along with other growth factors is essential in promoting the formation of new vessels, a process known as angiogenesis. In macrovessels, NO works to attenuate inflammation and cell adhesion. In particular, it inhibits thrombosis and promotes blood flow.

The three NOSs, as well as non-enzymatic processes, are also sources of endogenous NO in cardiomyocytes [91]. Within cardiomyocytes, the three NOS isoforms—nNOS, eNOS, and iNOS—have been described and are each expressed in typical positions. For example, nNOS is linked to the ryanodine receptor in the sarcoplasmic reticulum, and eNOS is found predominantly linked to caveolin-3 in the caveoles of the plasma membrane. In contrast, iNOS can be free in the cytoplasm, and it is generally induced in cardiomyocytes during inflammatory stimuli but also during ischemia or mechanical overload. Due to the unique spatial distribution and concentrations of the NO generator from the three NOS isoforms located in different domains, each NOS can affect the myocardium differently. In many situations, NO exerts a cardioprotective function, while in certain situations a cardio-damaging function has been described [92,93,94].

To maintain homeostasis of the NO/cGMP mechanism, cytoplasmic levels of cGMP are reduced by various phosphodiesterases, including those of types 5, 6, and 9 (PDE 5, 6, 9) [95]. Recent data suggest that thrombospondin-1 (TSP1) limits eNOS activity and NO signaling by directly targeting sGC or through direct inhibition of cGMP targets [96,97]. These limiting effects of TSP1 on the NO store signaling pathway interact with the vascular cell membrane receptor CD47 [98]. TSP1 is also emerging as a potential therapeutic target (see below).

### 3.2. NO in Myocardial Ischemia/Reperfusion Injury

A decrease in the formation and bioavailability of NO is a hallmark of several cardiovascular diseases. In particular, ischemia and reperfusion may result in reduced availability of NO. In reperfusion, this occurs together with reactive oxygen species (ROS) generation and Ca^2+^ overload, both leading to the prolonged opening of mitochondrial permeability transition pore (mPTP) and other processes leading to myocardial stunning, arrhythmias, cell death, and myocardial infarction. However, both detrimental and beneficial effects of NO have been described in many cardiovascular diseases. The potential confounding variables proposed by Schulz et al. [99] in 2004 are still relevant and may influence the experimental findings on the function of NO in myocardial I/R. These confounding factors include:*(a)* The lack of direct measures of myocardial NO concentration and/or NOS expression;*(b)* The lack of attention to non-enzymatic NO production as a potential source of NO;*(c)* The lack of consideration for plasma/blood components influencing NO delivery and metabolism.

Several papers have reported a detrimental role of NO in the I/R scenario [100,101,102,103,104,105,106,107,108]. Very often, iNOS-derived NO has been considered damaging. However, it plays an important role in several pathophysiological and physiological conditions, including coronary heart disease. Indeed, the role of iNOS/NO signaling in myocardial I/R injury is very complex, and both the beneficial and harmful effects of iNOS have been described. It is likely that a critical balance between NO and peroxynitrite (ONOO^−^, a strong oxidant derived from NO degradation that is often generated when there is an imbalance between NO and O_2_^−^ production) may participate in this dual role of iNOS. Notably, iNOS-derived NO was shown to play a cardioprotective role through its vasodilator and antioxidant effects in the so-called second window of protection [109], in which the antioxidant defense system may play a pivotal and prevalent role [110]. In particular, ischemic preconditioning prevents the subsequent overproduction of ONOO^−^ and, in doing so, protects the heart from oxidative damage. Therefore, the enhanced expression of antioxidants due to the elimination of oxidative stress leads to the iNOS switch from a deleterious enzyme to a protective one. In two elegant reviews, Bolli and collaborators evidenced that the majority of studies report the cardioprotective role of NO, including NO derived from iNOS [92,93]. Therefore, it is considered a strong mediator of cardioprotection (Figure 3).

### 3.3. NO as a Mediator of Cardioprotection

After the discovery of the preconditioning (PreC) and postconditioning (PostC) phenomena, reperfusion injury was recognized as a reality from which the heart must be protected. This is especially possible with PostC, which is under the control of doctors in cases of acute coronary disease. Several potentially cooperative and protective pathways, including RISK and SAFE, are activated by both PreC and PostC. In these pathways, phosphorylation/dephosphorylation reactions are widely represented, including phosphorylation/dephosphorylation of NOSs. However, cardioprotective modalities of signal transduction include the important contribution of redox signaling through ROS and S-nitrosylation by NO and derivatives such as nitroxyl (HNO) [111]. All of these protective modalities can interact and regulate multiple pathways, thus influencing each other.

Although NO derived by iNOS plays a central role in initiating and mediating the late phase or second window of ischemic preconditioning protection and the exogenous administration of NO prior to ischemia can initiate a preconditioning-like phenomenon, it has been proposed that endogenous NOS-derived NO is not involved in triggering or mediating the early phase of PreC protection [99]. However, endogenous NO is involved in PostC protection. Indeed, we reported that PostC protection can be abolished by the NOS-inhibitor L-nitro-arginine-methylester [112]. Subsequently NO produced by nNOS has been suggested as the mediator of PostC protection [113]. Endogenous NO is also involved in exercise-induced protection [114,115,116], as well as in cardioprotection induced by beta3-adrenoreceptor stimulation [114,117].

### 3.4. NO, Cardioprotection, and Mitochondria

The cardioprotective role of NO is certainly due to its interaction with the mitochondria (Figure 2). In fact, NO interacts with mitochondrial potassium channels, components of the electron transport chain, and/or the mPTP to limit myocardial damage from I/R. However, it is still not known which of these interactions is critical for protection [92,93,118]. Because of the linkage between NO/cGMP from one side and TIPS/CD47 from the other side, blocking or activating TIPS/CD47 signaling may reduce or exacerbate I/R injury, respectively [119,120]. In terms of protection, the NO–mitochondria axis promotes the main gain-of-function in post-ischemia cardiac tissues [113]. Particularly, NO-mediated cardioprotection, which maintains the functionality of mitochondria, may be due either to NO’s direct impact on mitochondria or to its indirect effect, which results in mitochondrial preservation [121]. As mentioned above, one mechanism by which NO exerts its cardioprotective effects is by interacting with mitochondrial K_ATP_ channels, which can reduce mitochondrial calcium overload and prevent the opening of mPTP. This can limit the extent of myocardial damage during I/R injury [122]. NO can also interact with components of the electron transport chain to reduce the production of ROS, which can cause oxidative damage to the mitochondria and surrounding tissue. For example, NO can inhibit the activity of complex I in the electron transport chain, which can reduce ROS production and limit oxidative damage [123]. In addition to its direct effects on mitochondria, NO can also modulate mitochondrial biogenesis and function through its interaction with cGMP. NO can activate sGC, which can increase cGMP levels and activate downstream signaling pathways that promote mitochondrial biogenesis and function [124]. Furthermore, one of the fundamental processes of ischaemic PreC and PostC seems to be that modest oxidative stress is able to activate antioxidant defense systems, which provide cardioprotection on an intermediate or long-term timescale. NO-dependent S-nitrosylation of metabolic/survival essential proteins in mitochondria may represent a more direct antioxidant mechanism implicated in NO-mediated PreC [125]. Consequently, these proteins are protected from irreversible thiol oxidation while still regulating mitochondrial oxygen consumption under hypoxic conditions (during the ischemic phase) [126]. Recent research has shown that hypoxic NO signaling in cardiomyocytes is linked to a post-translational mitochondrial complex I alteration that reduces oxidative stress [127]. Generally, the mechanism of NO’s cardioprotective effects is based on a balance between ROS/RNS formation and degradation, which may favor cardioprotective pathways, such as the NO-cGMP-PKG axis and optimal S-nitrosylation of proteins, or the stimulation of cardiotoxic pathways at high ROS/RNS levels. Chouchani and collaborators showed in vivo in mice that mitochondrial S-nitrosation at the reperfusion phase of myocardial infarction is cardioprotective using the mitochondria-selective S-nitrosating chemical MitoSNO. The S-nitrosation of mitochondrial complex I, which inhibits the reactivation of mitochondria during the vital first minutes of the reperfusion of ischemic tissue, was discovered to be responsible for protection by decreasing ROS generation, oxidative damage, and tissue necrosis [128]. Furthermore, by halting the progression of post-infarction heart failure, Metner et al. showed that mitochondria-targeted S-nitrosothiol (MitoSNO) provided acute reperfusion protection [129]. Overall, the literature suggests that NO plays a critical role in cardioprotection by interacting with the mitochondria through multiple mechanisms. Targeting the NO–mitochondria axis may, therefore, be a promising therapeutic approach for reducing myocardial damage during I/R injury.

### 3.5. NO and Clinical Applications

NO and drugs that release NO, which act directly or indirectly on NO-dependent signaling pathways, have been found and are being used for clinical applications both in age-related diseases and in younger patients (Table 1). As of today, one of the oldest drugs used to relieve angina is nitroglycerin (NTG), and it remains the treatment of choice for this purpose [130]. It has been hypothesized that NTG releases NO, which mediates most of its effects. Although this is true, many of the effects of NTG are attributed to the opening of potassium channels, and under certain circumstances, NTG might not act by releasing NO [131,132]. More recently, technologies that link NO donors to existing drugs to create hybrid drugs are in development, and some are under preclinical and clinical investigation. These include organic nitrites and nitrates, the so-called NONOates and NO-metal complexes, such as sodium nitroprusside [133,134]. These traditional methods to increase NO levels generally have limited clinical utility, owing to the development of tolerance and unfavorable pharmacokinetics. Several promising alternative therapeutic strategies have recently been proposed to increase NO signaling in the cardiovascular system. These alternative strategies aim to amplify the nitrate–nitrite–NO pathway or introduce new classes of drugs that donate NO and/or limit NO metabolism by modulating ROS and downstream phosphodiesterases and soluble cGMPs or identify new ways to improve NOS activity [135]. Other NO siblings and NO-related species such as HNO [136,137,138,139] and S-nitrothiols [34,140] have been tested with positive results in cell culture and animal studies related to cardiovascular function and diseases, but remain to be proven in clinical trials.

Therapeutic agents that block TSP1/CD47 signaling improve the sensitivity to NO-mediated vasodilators in cardiovascular diseases in humans [141] as well as in pigs and rodents, where these agents improve blood flow and hyperemia in response to ischemia [119,142,143].

In experimental and clinical studies within the cardiovascular field and outside it, donors of NO and drugs/strategies that target the NOS/NO system are also used. Noteworthy among these studies are those that use inhaled NO and the more recent studies that target the TIPS/CD47 system [144]. Recent clinical studies evaluating drugs targeting CD47 in oncology have confirmed the emerging interest in TIPS/CD47 in human diseases other than cardiovascular disease [145].

## 4. Carbon Monoxide (CO)

### 4.1. CO Biology in the Heart and Vessels

CO is a gaseous molecule that can easily diffuse through tissues. It is produced during heme degradation (Figure 2) by heme oxygenase (HO), which comes in two constitutive isoforms (HO-2 and HO-3) and an inducible form (HO-1) [146].

The enzymatic breakdown of heme into CO, biliverdin IX-α, and free Fe^2+^ requires oxygen and NAD(P)H as co-factors [147]. HO-1 functions as a defense mechanism against radical species produced during stressful stimuli such as I/R, hypoxia-reoxygenation, endo-toxic shock, carcinogenic radiations, and UV-A exposure [148,149]. CO shares similar biological activities with NO and, as for NO, can induce vasodilator effects by directly activating sGC [150]. CO is an endogenously synthesized gas (Figure 1) that has attracted the interest of cardiovascular researchers owing to its numerous physiological functions. Indeed, the recent literature has highlighted the importance of CO in the heart and vessels [151]. In the heart, CO has been shown to have cardioprotective effects (Figure 3). Studies have found that CO can reduce infarct size and improve cardiac function after I/R injury [152]. CO has also been shown to play a pivotal role in angiogenesis, with research demonstrating that it can promote vessel growth and improve blood flow. These effects are mediated through various signaling pathways, including the activation of HO-1 and NOS [153]. The same pathways can also regulate vasodilation [154,155]. Additionally, CO has been shown to have anti-inflammatory effects in the vessels, which can help to reduce the risk of atherosclerosis and other cardiovascular diseases [156].

### 4.2. CO in Myocardial Ischemia-Reperfusion Injury and Cardioprotection

HO-1 expression and activity increase during hypoxia but decline during reoxygenation, resulting in increased myocardial damage [149]. It was demonstrated that incubation of hypoxic cardiac myocytes with hemin or bilirubin can reduce reoxygenation-induced damage. The authors suggested that these protective effects are attributable to an increased bioavailability of the HO-1 substrate and the maintenance of high HO-1 activity during the reoxygenation phase [157]. The cardioprotective action of CO is related to its mitochondrial targeting (Figure 2), as demonstrated through the opening of the mitochondrial K_ATP_ channel and the inhibition of the opening of the mPTP [158]. CO also modulates several signaling pathways, including the NO/sGC pathway, mitogen-activated protein kinase (MAPK), and ROS [159]. CO donors (CO-RMs) have been used in preclinical and clinical studies, and their cardioprotective action has been evaluated in mouse models [160,161]. Male and female animals have common signaling pathways, but sex influences the expression of specific genes and proteins involved in the protection of mitochondrial and myocardial function, such as protein kinase B [161].

### 4.3. CO, Cardioprotection, and Mitochondria

Mitochondria are essential intracellular organelles involved in energy supply, and they are known targets of CO. Cellular metabolic adaptability, specifically, the balance between oxidative and glycolytic metabolism, is critical for cell survival in hypoxia and I/R diseases such as cerebral stroke, cardiac arrest, or myocardial infarction. CO has been shown to have protective effects on mitochondrial function at low doses, helping to maintain stable mitochondrial membrane potential and prevent alteration by pro-inflammatory stimuli. CO can modulate the production of mitochondrial reactive oxygen species (mtROS) in several cellular models [162]. It also regulates the activity of mitochondrial hemoproteins, including cytochrome c oxidase. Additionally, CO plays a role in the modulation of apoptosis [163,164] and inflammasome-dependent inflammation [165], both being mediated by soluble factors derived from mitochondria, such as cytochrome c and mitochondrial DNA (mtDNA) [162]. In a pig animal model of cardiac I/R, during coronary occlusion, the ratio of lactate generation over glucose intake was substantially decreased in CO-treated animals, demonstrating that CO administration preserves mitochondrial metabolism during the ischemic phase of the myocardium [166]. Consistently, CO-treated pigs that received cardiopulmonary bypass with cardioplegic arrest revealed considerably greater cardiac ATP and phosphocreatine levels at 1 h of reperfusion, suggesting that CO improves mitochondrial metabolism [167]. Moreover, CO has been shown to slow the progression of cardiomyopathy caused by doxorubicin by promoting mitochondrial biogenesis and enhancing cell metabolism [168]. Finally, in a rat fatal sepsis model, CO increased animal survival by boosting mitochondrial energy metabolism through the stimulation of mitochondrial biogenesis [169]. In summary, CO emerges as a molecule capable of enhancing the oxidative activity of mitochondrial metabolism in several experimental settings, including basic science cardiomyocyte models [170].

### 4.4. CO and Clinical Applications

CO has shown therapeutic potential as both an anti-inflammatory and cytoprotective agent in numerous organ/tissue injury and sepsis models, as reported by Ryter et al. [171]. Consistently, several therapeutic gases including CO have been tested to treat I/R injuries. This approach aims to treat ischemia-induced tissue damage and subsequent reperfusion exacerbations, such as those resulting from myocardial infarction, ischemic stroke, vascular atherosclerosis, transplant, severe malaria, severe sepsis, and autoimmune diseases [162]. Although some clinical studies have been initiated (Table 1), there is currently no solid data available from human studies. In animal models, novel cancer therapeutic strategies have been developed using free CO donors, such as [Ru(CO)_3_Cl_2_]_2_ and Mn_2_(CO)_10_, as well as CO gas administration [172]. Despite the growing understanding of the importance of CO in cardiovascular biology, there is still much that is unknown about its role in the heart and vessels. Future research could help to elucidate the specific signaling pathways through which CO exerts its effects, as well as identify potential therapeutic targets for cardiovascular disease.

## 5. Gasotransmitters Interplay within the Cardiovascular System

Important insights about the role of gasotransmitters in the vascular system arise from a very recent review from Juin *and colleagues* [173] focused on diabetic nephropathy, a major risk factor for cardiovascular complications, typically characterized by endothelial dysfunction and altered gasotransmitter production. Different diabetic experimental models showed an association between lower CO, NO, and H_2_S (and their relative biosynthetic enzymes) levels and endothelial damage. Deletion of HO-1/HO-2 or eNOS in diabetic murine models worsened the outcomes in terms of vascular complications, while the use of NO/H_2_S donors or HO/eNOS inducers was shown to be effective in mitigating renal injury. The mechanisms proposed to be involved in the gasotransmitter-mediated protection included the activation of N-methyl-D-aspartate (NMDA) and peroxisome proliferator-activated receptor-γ (PPAR-γ) receptors, the regulation of extracellular matrix proteins and matrix metallo-proteinases (MMPs), and the expression of gap junctions and others integral membrane proteins [173].

Interestingly, all three gases cooperate (and sometimes overlap) in the regulation of these mechanisms. Indeed, as discussed in the previous sections, H_2_S, NO, and CO share their physiological actions. All three gasotransmitters are able to induce vasodilation, inhibit apoptosis, regulate mitochondrial function, and reduce oxidative stress and inflammation (Figure 3). Moreover, it seems that one gasotransmitter may influence the others in terms of production and signaling, thus resulting in synergistic (or antagonistic) effects. For example, Coletta et al. [174] demonstrated that H_2_S and NO have a mutual relationship in regulating angiogenesis and vasorelaxation. In particular, they observed that the inhibition of CSE or NO synthase abolishes, respectively, NO- and H_2_S-mediated angiogenesis/vasodilation. One explanation, which was also suggested by the authors, may be a joined action on a common downstream target, namely, cGMP: while NO activates sGC to produce cGMP, concomitantly, H_2_S inhibits PDE5, slowing down cGMP degradation [174,175]. H_2_S-NO cooperation was also demonstrated by other research groups. In eNOS KO mice experimentally subjected to cardiopulmonary arrest, exogenous sulfide administration failed to induce cytoprotection [176], while in CSE KO mice, a worse myocardial I/R injury was demonstrated to be related to lower eNOS phosphorylation and NO bio-availability [177]. Another research team demonstrated that exogenous sulfide therapy was effective in protecting against murine heart failure, and this effect was eNOS/NO-dependent [178]. Furthermore, a study using CSE KO mice subjected to femoral artery ligation observed that H_2_S regulates ischemic vascular remodeling in a NO-dependent fashion. In the same study, the researchers found that there was a defined temporal relationship in the induction of the two gases synthesis. Specifically, H_2_S production initiates upon limb ischemia, followed by an increase in NO bioavailability. However, as NO plasma levels increase, H_2_S concentration is restored to baseline. These data strengthen the concept of a highly regulated crosstalk between the gasotransmitters to maintain homeostasis [179]. Regulation of eNOS by H_2_S has been studied in different models. It has been proposed that sulfide-dependent eNOS activation could be mediated by the PIP3/Akt pathway. A higher Akt activation enhances eNOS phosphorylation at S1177 and/or S1179, thus resulting in an increase in NO-synthesizing activity. Another proposed mechanism by which sulfide regulates eNOS is direct persulfidation at C443 [180]. However, as previously mentioned, cooperation between H_2_S and NO may result in antagonistic behavior. For example, S-nitrosylation at C150 of GAPDH leads to the blunting of its catalytic activity, while S-sulfhydration at the same residue increases the enzyme activity [181]. In addition to the biological interaction between NO and H_2_S, the literature describes chemical interactions as well. These interactions may result in the production of nitrosopersulfide (ONSS^−^), thionitrous acid (HSNO), polysulfides (HS_n_^−^), and dinitrososulfite (SULFI/NO), the first being the strongest NO releaser [180,182]. HSNO can further react with H_2_S forming S-nitrosothiol (RSNO) and hydrogen disulfide (HSSH) [183]. According to Filipovic et al., the smallest S-nitrosothiol, HSNO, is created when H_2_S combines with S-nitrosothiols. They proved that HSNO can be processed to produce NO^+^, NO, and NO species at the cellular level. The transnitrosation of proteins like hemoglobin is made possible by the fact that HSNO may freely diffuse through membranes. To recap, they provided an explanation for some of the physiological effects attributed to H_2_S while also introducing HSNO as a new messenger molecule and indicating that it may be crucial for cellular redox regulation [17,184]. Regarding CO-NO communication, we previously mentioned that they share a signaling pathway involving the activation of sGC to induce vasodilation (see Section 3.1 and Section 4.1). Moreover, it has been demonstrated that CO is able to regulate NO production in both a negative and a positive fashion, depending on its concentration, in endothelial and immune cells. In particular, physiological levels of CO result in increased NO levels via the accumulation of an intracellular pool of CO-displaceable NO [185]. In an in vitro model of human umbilical vein ECs (HUVECs), treatment with a CO-RM or an HO-1 inducer was able to rescue TNF-α-dependent downregulation of eNOS, and this effect was eliminated in the presence of HO-1 inhibitor. Conversely, it has been reported that high levels of CO can inhibit the catalytic activity of eNOS in vitro, by binding its prosthetic heme moiety. Moreover, NAD(P)H is required by both HO-1 and NOSs as a reducing factor, suggesting that a hyper-induction of HO-1 may lead to a competition with eNOS for NAD(P)H, thus lowering endothelial NO synthesis. However, the relationship between NO and CO is mutual, since it has been observed that NO itself and its RNS by-products are potent inducers of HO-1 and, thus, are able to increase CO levels [186]. The body of evidence also demonstrated an inter-talk between H_2_S and CO. H_2_S seems to be able to induce the expression of HO-1, thereby determining an increase in CO. The proposed mechanism involves the persulfidation of Kelch-like ECH-associated protein 1 (Keap1) C151, which determines the dissociation of the protein from nuclear factor (erythroid-derived 2)-like 2 (Nrf2). Translocation of Nrf2 into the nucleus lastly results in a higher transcription of HO-1. Both CO and NO, on their own, are able to reversibly bind the noncatalytic heme group of CBS, and this aspect was observed in different cell types and tissues, including the vasculature. This inhibition has been reported to lower the H_2_S levels. However, recent studies suggest that it may, instead, result in overall increased production, and this could be related to the fact that inhibiting CBS may switch the transsulfuration pathway to H_2_S production mediated by CSE, which has a higher bio-synthetic activity than CBS [180]. Moreover, CO and NO can also modulate H_2_S bioavailability by the inhibition of cytochrome c oxidase and the subsequent inhibition of the mitochondrial sulfide oxidizing pathway [187,188]. We previously discussed how CO, NO, and H_2_S activity converges in the elevation of cGMP to induce vasodilation. Another example of convergence happens in vascular smooth muscle cells: CO and NO enhance the opening of big-conductance K_Ca_ channels, while H_2_S activates different K_Ca_ channels and K_ATP_ channels, thereby decreasing the overall intracellular free calcium content [174,181,183].

In light of what is discussed above, a multi-target therapeutic approach that involves the exogenous administration of a combination of the three gasotransmitters is noteworthy. It was demonstrated that a hybrid nanoparticle system (mPEG-PLGH integrated with diethylenetriamine NONOate and thiobenzamide), engineered to release both H_2_S and NO, was more effective in inducing tube formation and sprouting new vessels than their single-coated counterparts in HUVECs. Similarly, the usage of ZYZ-803, another hybrid NO/H_2_S releaser, induced a stronger pro-angiogenic effect in rat aortic rings than the relative single-moiety molecules [189].

## 6. Conclusions

Gasotransmitters are among the chemicals that have been convincingly shown to be essential in initiating ischemic PreC [111,190,191] and modulating PostC effects [125,192,193]. Myocardial infarction, which accounts for 46% of all major adverse cardiac events in men and 38% in women globally [194], continues to be the most common cause of death and morbidity worldwide. The fundamental challenge in treating myocardial loss following infarcting ischemia is reducing the so-called reperfusion injury to minimize ischemia/hypoxia-driven cell death, which can occur via intracellular Ca^2+^ overload, oxidative stress, apoptosis and necrosis, and subsequent inflammation [195]. To better protect the myocardium, many intracellular pathways that have been shown to play a role in both the pre- and post-ischemic phases of I/R damage can be targeted. In this regard, a number of treatment strategies to protect post-ischemic myocardium have been suggested. After years of intensive basic research on the subject, in vivo investigations and ongoing clinical trials have suggested potential clinical relevance for the use of exogenous gasotransmitters and for the control of their endogenous production. Moreover, the significance of gasotransmitter interactions has been proposed as a potential treatment approach. Gasotransmitters are not an exception to the synergistic effects of complex signaling pathways in providing cardioprotection. For instance, intricate interactions between H_2_S, CO, and NO have been shown in numerous experimental models [26,183], and a number of substrates are subjected to post-transcriptional regulation through the nitrosylation/persulfidation of certain residues. It is clear that mitochondria are involved in the majority of cardioprotective signaling pathways, and research has shown that post-translational nitrosylation and persulfidation specifically target a number of mitochondrial components [196,197]. Yet, we should not forget that for optimal cardioprotection, combining additive or synergistic multitarget therapy may be required [198].

## Figures and Tables

**Figure 1 ijms-24-12480-f001:**
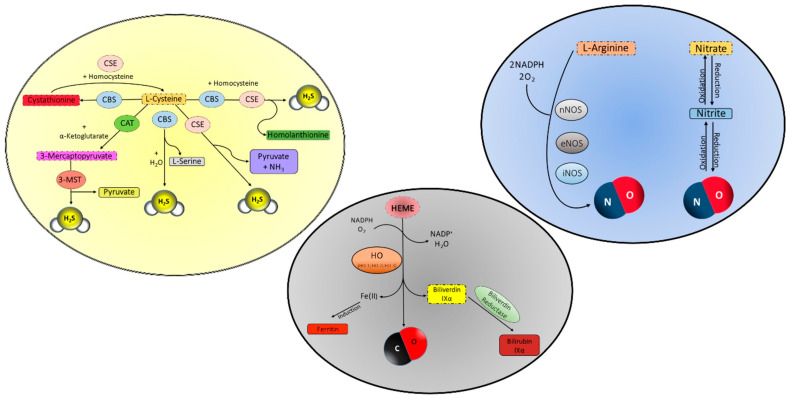
Biosynthesis of the three gasotransmitters. H_2_S is synthesized by cystathionine β-synthase (CBS) or cystathionine γ-lyase (CSE) by converting L-cysteine to L-serine or pyruvate and NH_3_. L-cysteine can also be metabolized in 3-Mercaptopyruvate via cysteine aminotransferase (CAT) and then converted into H_2_S and pyruvate by mercaptopyruvate transferase (3MST); CO is synthesized by heme oxygenases (HO-1, HO-2, or HO-3) by oxidizing heme with the help of cofactors, such as NADPH and O_2_, to produce CO, biliverdin, and ferritin; NO is synthesized by a family of NO synthases (nNOS, eNOS, or iNOS) that catalyzes the oxidation of L-arginine to L-citrulline producing NO in the process. O_2_ and NADPH are necessary co-factors.

**Figure 2 ijms-24-12480-f002:**
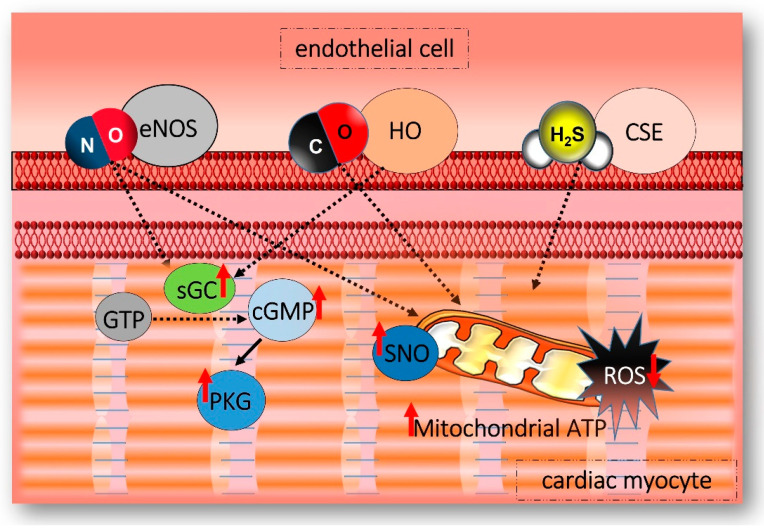
Main targets of the three gasotransmitters in a cardiovascular *scenario*. NO, H_2_S, and CO target mitochondria with multiple effects. NO favors S-nitrosylation (SNO) on mitochondrial proteins and also increases the expression of soluble guanylate cyclase (sGC), activating the related pathways with the aim to inhibit cardiac hypertrophy. NO also induces vasorelaxation through protein kinase G (PKG) activation. CO increases ATP production by targeting mitochondria, and it is able to regulate the NO/sGC pathway, whereas H_2_S has a scavenger function, which reduces ROS and inhibits apoptosis.

**Figure 3 ijms-24-12480-f003:**
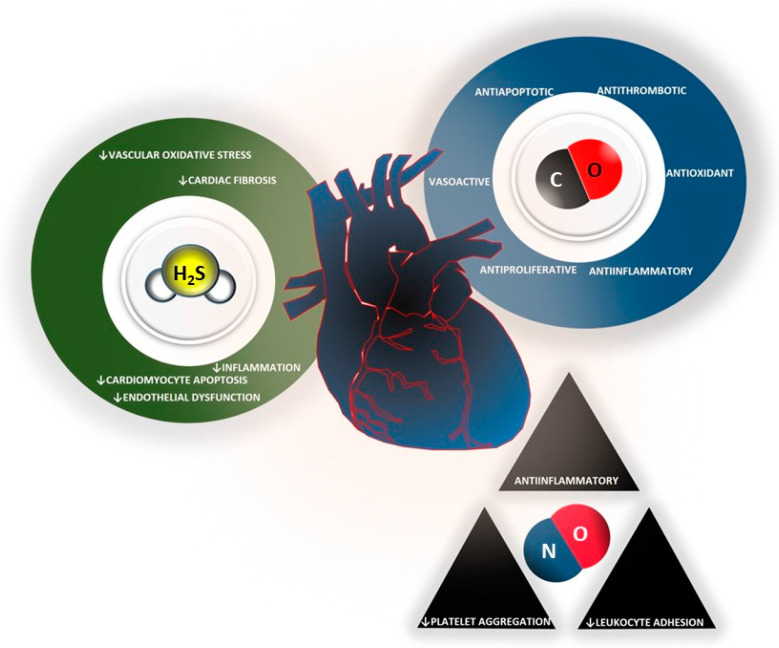
Cardiac effects of the three gasotransmitters. H_2_S, by scavenging ROS, reduces vascular oxidative stress and so reduces inflammation, cardiac fibrosis, cardiomyocyte apoptosis, and endothelial dysfunction. NO has anti-inflammatory effects related to the inhibition of platelet aggregation and leukocyte adhesion. CO has anti-inflammatory and anti-proliferative activity. In addition, CO has a vasoactive, anti-apoptotic, anti-thrombotic, and antioxidant function.

**Table 1 ijms-24-12480-t001:** The three gasotransmitters in clinical trials. The clinical trials registered at *ClinicalTrials.gov* involve the administration of one of the gasotransmitters as an intervention, with a particular focus on those related to cardiovascular diseases/effects.

Study Title	Condition	Intervention	Outcome(s)	Location(s)	Trial ID
*Clinical trials with H_2_S donors*					
A Double-Blind, Controlled Study to Compare the Gastrointestinal Safety of a 14-Day Oral Dosing Regimen of ATB-346 to Sodium Naproxen in Healthy Subjects	Gastric ulcer	ATB-346 (H_2_S-NSAID) vs. naproxen sodium (NSAID)	Gastroduodenal ulcers ≥ 3 mm diameter (Time frame: after 14 days of oral dosing) Gastroduodenal ulcers ≥ 5 mm diameter; gastroduodenal erosions; dyspepsia; hematocrit; thromboxane B2 levels (Time frame: after 14 days of oral dosing)	Topstone Clinical Research Toronto, Ontario, Canada, M9C 4Z5	NCT03291418
Groningen Intervention Study for the Preservation of Cardiac Function With Sodium Thiosulfate After ST-segment Elevation Myocardial Infarction *	Myocardial infarction HF	sodium thiosulfate vs. placebo	Myocardial infarct size measured with late gadolinium enhancement cardiac MRI (Time frame: 4 months after randomization) LV ejection fraction as assessed with cardiac MRI; NT-proBNP level (ng/L) (Time frame: 4 months after randomization) All-cause mortality; combined major CV-AEs; incidence of stroke; incidence of stent thrombosis; incidence of ICD implantation; hospitalization for HF or chest pain (Time frame: 4 months after randomization and after 2-year follow-up) Enzymatic infarct size as assessed with peak CK-MB (U/L) (Time frame: 0–3 days after randomization) Health related quality of life: EuroQol EQ-5D-5L; general affective status (Time frame: 0–5 days after randomization and at 4 months follow up) CK (U/L); troponin T (ng/mL) (Time frame: 0–3 days after randomization) NT-proBNP (ng/L) (Time frame: 0–5 days after randomization)	Treant Scheper Hospital Emmen, Drenthe, Netherlands, 7824 AA University Medical Centre Groningen Groningen, Netherlands, 9700RB University Medical Center Utrecht Utrecht, Netherlands, 3584 CX	NCT02899364
A Dose Escalation Study to Assess the Safety and Ability of SG1002 to Overcome Circulating Deficits in Hydrogen Sulfide Found in Heart Failure	HF	SG1002 (α-sulfur/sodium sulfate) vs. placebo	Number of subjects with AEs (Time frame: following 7 days of treatment at each dose) Changes in peak H_2_S levels in HF subjects following SG1002 administration (Time frame: 24 h) Potential clinical benefits of SG1002 administration by analyzing BNP levels (Time frame: 7 days at each dose)	Alfred Health Melbourne, Victoria, Australia, 3004 Nucleus Network Melbourne, Victoria, Australia, 3004	NCT01989208
Patients Peripheral Vascular Effects of Sulfhydryl-containing Antihypertensive Pharmacotherapy on Microvascular Function and Vessel Remodeling in Hypertensive Humans *	Hypertension	Captopril (ACEi+SH) vs. enalapril (ACEi) vs. hydrochlorothiazide (diuretic)	Laser Doppler blood flow (Time frame: 16 weeks) SBP; DBP (Time frame: 16 Weeks)	Pennsylvania State University University Park, Pennsylvania, United States, 16802	NCT03179163
**Abbreviations**: HF = heart failure; MRI = magnetic resonance imaging; LV = left centricular; NTproBNP = N-terminal pro type B natriuretic peptide; CV-AEs = cardiovascular adverse events; ICD = implantable cardioverter-defibrillator; CK-MB = creatin kinase-muscle brain; CK = creatin kinase; ACEi = ACE inhibitor; SBP = systolic blood pressure; DBP = diastolic blood pressure; AEs: adverse events; BNP = B type natriuretic peptide * = ongoing studies					
*Clinical trials with NO donors/NO potentiators*					
Effects of Nitric Oxide on Vascular Responsiveness and on Endothelial Cells During Hemolysis in Patients With Pre-operative Endothelial Dysfunction Undergoing Prolonged Cardiopulmonary Bypass *	Endothelial dysfunction Intravascular hemolysis	iNO vs. placebo	RHI (Time frame: perioperatively—before anesthesia induction—and at 24 h after CPB during ICU admission) eNOS enzymatic activity (Time frame: perioperatively—before anesthesia induction—and at 24 h after CPB during ICU admission) PVR; SVR (Time frame: every 6 h after surgery for 24 h after CPB start)	Massachusetts General Hospital Boston, Massachusetts, United States, 02114 Boston Medical Center Boston, Massachusetts, United States, 02118	NCT03748082
Effects of Prolonged Delivery of Nitric Oxide Gas on Plasma Reduction-Oxidation Reactions in Cardiac Surgical Patients *	Endothelial dysfunction	iNO vs. placebo	Changes in the concentration and electric potential of GSH/GSSG and Cys/CysSS couples in the plasma after cardiac surgery (Time frame: before cardiac surgery and during the first 48 h aftersurgery) Changes in concentration of plasma and RBCs of NO metabolites after cardiac surgery (Time frame: before cardiac surgery and after the first 48 h after surgery)	Massachusetts General Hospital Boston, Massachusetts, United States, 02114 Southampton General Hospital Southampton, Hampshire, United Kingdom, SO16 6YD	NCT04022161
Improving Outcomes in Cardiac Arrest With Inhaled Nitric Oxide *	Cardiac arrest	iNO	Rate of return of spontaneous circulation; change in cerebral oxygenation (Time frame: 1 day) Neurologic outcomes at hospital discharge; short term survival (Time frame: up to 24 weeks)	Stony Brook University S. Setauket, New York, United States, 11720	NCT04134078
Nebivolol and the Endothelin (ET)-1 System	Prehypertension Hypertension	nebivolol (β-blocker with NO-potentiating properties) vs. metroprolol (β-blocker) vs. placebo	SBP; DBP (Time frame: before and after intervention) Percent change in FBF response to BQ-123 (100 Nmol/Min) (Time Frame: 0–60 min before and after the intervention) Percent change in FBF response to BQ-123 (100 Nmol/Min) + BQ-788 (50 Nmol/Min) (Time frame: 0–120 min before and after the intervention) FBF response to ACh or sodium nitroprusside; FBF response to ACh w/ or w/o BQ-123+BQ-788 (Time frame: before and after the intervention)	UC-Boulder Clinical and Translational Research Center Boulder, Colorado, United States, 80309	NCT01395329
Acute Effects of Inhaled Sodium Nitrite on Cardiovascular Hemodynamics in Heart Failure With Preserved Ejection Fraction	HF	inhaled sodium nitrite vs. placebo	Change in pulmonary capillary wedge pressure (mmHg) during exercise (Time frame: baseline, after study drug dosing, approximately 4 min after starting exercise)	Mayo Clinic Rochester, Minnesota, United States, 55905	NCT02262078
A Randomized, Double-blinded, Placebo-controlled, Phase IIa Dose-ranging Study to Assess the Safety, Pharmacokinetics, and Tolerability of Multiple Doses of Sodium Nitrite in Patients With Peripheral Arterial Disease (PAD)—SONIC	Peripheral arterial disease	80 mg sodium nitrite vs. 40 mg sodium nitrite vs. placebo	Reporting of AEs during 11 week treatment period (Time frame: 11 weeks) Assessment of changes in brachial artery FMD 10 weeks after baseline (Time frame: 10 weeks) Assessment of changes in walking distance (Timeframe: 10 weeks) Assessment of improvement in quality of life using the WIQ and SF-36 Questionnaire (Time frame: 10 weeks)	University of Colorado Denver Health Sciences Center Aurora, Colorado, United States, 80045 Emory University Atlanta, Georgia, United States, 30322 University of Iowa Iowa City, Iowa, United States, 52242 University of Cincinnati Cincinnati, Ohio, United States, 45267 Cleveland Clinic Cleveland, Ohio, United States, 44106 Ohio State University Columbus, Ohio, United States, 43210 University of Pennsylvania Philadelphia, Pennsylvania, United States, 19104 Vanderbilt Heart and Vascular Institute Nashville, Tennessee, United States, 37232 Medical College of Wisconsin Milwaukee, Wisconsin, United States, 53226	NCT01401517
Efficacy of Oral Sodium Nitrite for Improving Physiological Functions in Older Adults	Aging	sodium nitrite vs. placebo	Vascular function (Time frame: 3 months) Motor function (Time frame: 3 months) Systemic oxidative stress and inflammation (Time frame: 3 months) Number of participants with additional measures of motor ability (Time frame: 3 months) Endothelial cell oxidative stress and inflammation (Time frame: 3 months) Plasma metabolites (Time frame: 3 months)	Clinical Translational Center Boulder, Colorado, United States, 80309	NCT02393742
The Effects of Inorganic Nitrate on Cardiac Muscle: Physiology, Pharmacology and Therapeutic Potential in Patients Suffering From Angina	Chronic stable angina	sodium nitrate vs. placebo	Time to 1 mm ST depression (Time frame: 12 weeks) Onset of chest pain (Time frame: 12 weeks) Change in TDI systolic peak velocity (Time frame: 12 weeks) Angina frequency (Time frame: 12 weeks) Nitrate/nitrite in plasma (Time frame: 12 weeks) Metabolic, inflammatory, and angiogenic plasma markers (Time frame: 12 weeks)	Cardiovascular Research Facility, University of Aberdeen Aberdeen, United Kingdom, AB24 3FX	NCT02078921
**Abbreviations**: AEs = adverse events; iNO = inhaled NO; RHI = reactive hyperaemia index; CPB = cardiopulmonary bypass; ICU = intensive care unit; RBCs = red blood cells; SBP = systolic blood pressure; DBP = diastolic blood pressure; FMD = flow-mediated dilation; FBF = forearm blood flow; ACh = acetyl choline; WIQ = Walking Impairment Questionnaire; TDI = tissue doppler imaging * = ongoing studies					
*Clinical trials with CO donors*					
Modification of Chronic Inflammation by Inhaled Carbon Monoxide in Patients With Stable COPD	COPD	iCO vs. placebo	Percentage of neutrophils in induced sputum (Time frame: 17 h after the last inhalation) Methacholine provocation threshold; exhaled CO/NO; FEV_1_, FVC, R_AW_, sG_AW_; inflammatory parameters in sputum and blood; 8-isoprostane in exhaled breath (Time frame: 17 h after the last inhalation)	University Medical Center Groningen, Department of Pulmonary Diseases Groningen, Netherlands, 9700RB	NCT00122694
**Abbreviations**: iCO = inhaled CO; FEV_1_ = forced exhaled volume in 1 s; FVC = forced vital capacity; R_AW_ = airway resistance; sG_AW_ = specific airway conductance					

**Table 2 ijms-24-12480-t002:** NOSs regulation. The table summarizes the variety of NOSs regulation mechanisms. eNOS, the most characterized, and nNOS are regulated in a similar fashion (allosteric modulation, phosphorylation, non-enzymatic processes). iNOS, which is the solely inducible isoform, is constitutively active once expressed. The relative cellular localization of NOSs isoforms is also listed.

NOS Isoform	Cellular Localization	Modulation			
**eNOS**	Plasma membrane (linked to caveolin)	**Enzymatic Regulation (phosphorylation)**			
		**Enzyme**	**P-site**	**Stimuli**	**Effect**
		Akt	S615	Shear stress; VEGF; statins; BK; 8-Br-cAMP	↑ activity
		PKA; Pim1	S633	?	
		Akt1; AMPK; PKA; CaMKII	S1177	Estrogens; VEGF; IGF-1; insulin; BK; shear stress; 8-Br-cAMP; statins; leptin; adiponectin; sphingosine 1-P	
		PKC	T495 (constitutively phosphorylated)	-	↓ activity
		PYK2	Y657	?	↑ activity
		**Non-enzymatic regulation**			
		**Trigger**	**Mechanism**		**Effect**
		↑ iCa^2+^	↑ CaM binding		↑ activity
		Allosteric regulation			
		Molecule	Mechanism		Effect
		HSP90	Dimer stabilization		↑ activity
iNOS	Cytoplasm	**Transcriptional regulation**			
		**Trigger**	**Mechanism**	**Stimuli**	**Effect**
		LPS	NF-κB	Inflammation; infections; ischemia; mechanical overload	↑ expression
		IL-1β			
		INF-γ	JAK-STAT		
nNOS	Sarcoplasmic reticulum (linked to ryanodine receptor)	**Enzymatic regulation (phosphorylation)**			
		**Enzyme**	**P-site**		**Effect**
		CaMKI	S741		↓ activity
		CaMKII	S852		
		?	S1212		↑ activity
		**Non-enzymatic regulation**			
		**Trigger**	**Mechanism**		**Effect**
		↑ iCa^2+^	↑ CaM binding		↑ activity
		**Allosteric regulation**			
		**Molecule**	**Mechanism**		**Effect**
		PIN	Dimer destabilization		↓ activity

Abbreviations: VEGF: vascular endothelial growth factor; BK: bradykinin; 8-Br-cAMP: 8-bromo-adenosine-3′,5′-cyclic monophosphate; PKA: protein kinase A; Pim1: proviral integration site for Moloney murine leukemia virus-1; AMPK: AMP-activated protein kinase; CaMKII: Ca^2+^/calmodulin-dependent kinase II; IGF-1: insulin-like growth factor-1; PKC: protein kinase C; PYK2: protein-tyrosine kinase 2-β; iCa^2+^: intracellular Ca^2+^; CaM: Ca^2+^/calmodulin; HSP90: heat shock protein 90; LPS: lipopolysaccharide; NF-κB: nuclear factor kappa-light-chain-enhancer of activated B cells; IL-1β: interleukin-1β; INF-γ: interferon-γ; JAK: Janus kinase; STAT: signal transducer and activator of transcription protein; CaMKI: Ca^2+^/calmodulin-dependent kinase I; PIN: protein inhibitor of nNOS.

## Data Availability

Not applicable.

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
