# Peer review of "Clinical Applications for Gasotransmitters in the Cardiovascular System: Are We There Yet?"

_ijms, 2023, doi:10.3390/ijms241512480_

Round 1

Reviewer 1 Report

The manuscript entitled: Clinical Applications for Gasotransmitters in the Cardiovascular System: are we there yet?, by Arrigo et al., presents a comprehensive and concise overview of the current literature, highlighting the pivotal role of gasotransmitters in the cardiovascular system. The article effectively explores the biosynthesis, mechanisms of action, and therapeutic effects of three main gasotransmitters.

Additionally, it provides valuable insights and precise data on their clinical applications, including pertinent studies on ischemia-reperfusion injuries, a prominent cardiovascular pathology.

To further enhance the depth and breadth of the individual chapters and elevate the overall quality of the manuscript, authors are strongly advised to consider the following suggestions:

Sentence 28: steric hindrance and charge

Sentence 45-55: References are missing? 

Sentence 60: Between 28 -34 Da?

Sentence 108: a constitutive H2S producer, replace with a constitutively expressed H2S producing enzyme. 

Sentence 117: Include and comment about the role of TRPa1 channels in this section: 

Eberhardt, M., Dux, M., Namer, B. et al. H2S and NO cooperatively regulate vascular tone by activating a neuroendocrine HNO–TRPA1–CGRP signalling pathway. Nat Commun 5, 4381 (2014). https://doi.org/10.1038/ncomms5381

Sentence 155: Include and comment the latest work related to the role of exogenous H2S in protecting the heart against I/R injury:

Add additional comment related to the similar chemical-biology approaches from the literature if possible.

Miljkovic, J. L., Burger, N., Gawel, J. M., Mulvey, J. F., Norman, A. A., Nishimura, T., ... & Murphy, M. P. (2022). Rapid and selective generation of H2S within mitochondria protects against cardiac ischemia-reperfusion injury. Redox Biology, 55.

Sentence 160: Include this reference: 

Chemical Biology of H2S Signaling through Persulfidation. Milos R. Filipovic, Jasmina Zivanovic, Beatriz Alvarez, and Ruma Banerjee. Chemical Reviews 2018 118 (3), 1253-1337

DOI: 10.1021/acs.chemrev.7b00205

Sentence 168-170: Add comment on the absence of the appropriate and reliable method for sensitive and selective determination of H2S and H2S related protein PTMs. 

Sentence 176: Include the references: 

Vitvitsky V, Miljkovic JL, Bostelaar T, Adhikari B, Yadav PK, Steiger AK, Torregrossa R, Pluth MD, Whiteman M, Banerjee R, Filipovic MR. Cytochrome c Reduction by H2S Potentiates Sulfide Signaling. ACS Chem Biol. 2018 Aug 17;13(8):2300-2307. doi: 10.1021/acschembio.8b00463. Epub 2018 Jul 18. PMID: 29966080; PMCID: PMC6450078.

Domán A, Dóka É, Garai D, Bogdándi V, Balla G, Balla J, Nagy P. Interactions of reactive sulfur species with metalloproteins. Redox Biol. 2023 Apr;60:102617. doi: 10.1016/j.redox.2023.102617. Epub 2023 Jan 27. PMID: 36738685; PMCID: PMC9926313.

Sentence 189: SQOR is bond to the inner membrane of mitochondria where interacts with CoQ pool that is essential for its activity and acts as a regulator of H2S signaling, discriminating between physiological and poisoning outcomes.

Sentence 191: Initially, SQOR consumes H2S and oxidised CoQ to transfer the thiol moiety to the reduced GSH thus forming GSSH (glutathione persulfide) and reduced CoQH2.  Alternatively, SQR can use sulfite (SO3) instead of GSH to form thiosulfate (S2O3). Glutathione persulfide is processed via ETHE1 to form sulfite (SO3) which participate in further H2S related catabolic processes in mitochondria. 

Consider consulting:

Organization of the Human Mitochondrial Hydrogen Sulfide Oxidation Pathway. Libiad, Marouane et al., Journal of Biological Chemistry, Volume 289, Issue 45, 30901 - 30910

Sentence 210-215:  Add and comment:

Miljkovic, J. L., Burger, N., Gawel, J. M., Mulvey, J. F., Norman, A. A., Nishimura, T., ... & Murphy, M. P. (2022). Rapid and selective generation of H2S within mitochondria protects against cardiac ischemia-reperfusion injury. Redox Biology, 55.

Sentence: 237: Homocysteinemia-related phenotype should be considered when using the CBS knockout cell model systems? 

Sentence 251: Hibernation or suspended like animated state?

Sentence 457-461: Add references and comments: 

Chouchani, E., Methner, C., Nadtochiy, S. et al. Cardioprotection by S-nitrosation of a cysteine switch on mitochondrial complex I. Nat Med 19, 753–759 (2013). https://doi.org/10.1038/nm.3212

Methner C, Chouchani ET, Buonincontri G, Pell VR, Sawiak SJ, Murphy MP, Krieg T. Mitochondria selective S-nitrosation by mitochondria-targeted S-nitrosothiol protects against post-infarct heart failure in mouse hearts. Eur J Heart Fail. 2014 Jul;16(7):712-7. doi: 10.1002/ejhf.100. Epub 2014 May 31. PMID: 24891297; PMCID: PMC4231226.

Sentence 629-631: Include the references: 

Chemical Characterization of the Smallest S-Nitrosothiol, HSNO; Cellular Cross-talk of H2S and S-Nitrosothiols. Milos R. Filipovic, Jan Lj. Miljkovic, Thomas Nauser, Maksim Royzen, Katharina Klos, Tatyana Shubina, Willem H. Koppenol, Stephen J. Lippard, and Ivana Ivanović-Burmazović

Journal of the American Chemical Society 2012 134 (29), 12016-12027

DOI: 10.1021/ja3009693

Eberhardt, M., Dux, M., Namer, B. et al. H2S and NO cooperatively regulate vascular tone by activating a neuroendocrine HNO–TRPA1–CGRP signalling pathway. Nat Commun 5, 4381 (2014). https://doi.org/10.1038/ncomms5381

Any additional comments on HNO and HNO related donors as well as their mechanism in respect of vasodilatation or ionotropic effect, from the literature are welcome. 

Author Response

Reviewer #1

Arrigo et al. reviewed the knowledge so far accumulated about gasotransmitters and their possible therapeutic application in the field of cardiovascular diseases. I report hereby comments:

The manuscript entitled: Clinical Applications for Gasotransmitters in the Cardiovascular System: are we there yet? by Arrigo et al., presents a comprehensive and concise overview of the current literature, highlighting the pivotal role of gasotransmitters in the cardiovascular system. The article effectively explores the biosynthesis, mechanisms of action, and therapeutic effects of three main gasotransmitters. Additionally, it provides valuable insights and precise data on their clinical applications, including pertinent studies on ischemia-reperfusion injuries, a prominent cardiovascular pathology.

To further enhance the depth and breadth of the individual chapters and elevate the overall quality of the manuscript, authors are strongly advised to consider the following suggestions:

Dear reviewer, we would like to thank you for the careful revision you performed on our manuscript; we strongly believe that the overall quality of our work will be strengthened by your suggestions.

Here it is a detailed list of corrections we did upon your comments:

- Sentence 28: steric hindrance and charge

Thank you for the suggestion, changes have been made accordingly

- Sentence 45-55: References are missing?

Thank you for your request to include a bibliographic citation for the current passage in our review. However, we believe that doing so would not be necessary in this case. The sentence in question represents a general reflection on the topic, rather than a specific reference to a particular source. As such, we feel that adding a bibliographic citation could be misleading, as it could imply that the idea expressed in the sentence is exclusively attributable to a single source. We hope this explanation clarifies our position on the matter.

- Sentence 60: Between 28 -34 Da?

Thank you for noticing it, correction has been done.

- Sentence 108: a constitutive H2S producer, replace with a constitutively expressed H2S producing enzyme.

The sentence has been changed accordingly.

- Sentence 117: Include and comment about the role of TRPa1 channels in this section:

Eberhardt, M., Dux, M., Namer, B. et al. H2S and NO cooperatively regulate vascular tone by activating a neuroendocrine HNO–TRPA1–CGRP signalling pathway. Nat Commun 5, 4381 (2014). https://doi.org/10.1038/ncomms5381

Thank you for your suggestion. A brief discussion has been added in lines 118-127.

- Sentence 155: Include and comment the latest work related to the role of exogenous H2S in protecting the heart against I/R injury:

Add additional comment related to the similar chemical-biology approaches from the literature if possible. Miljkovic, J. L., Burger, N., Gawel, J. M., Mulvey, J. F., Norman, A. A., Nishimura, T., ... & Murphy, M. P. (2022). Rapid and selective generation of H2S within mitochondria protects against cardiac ischemia-reperfusion injury. Redox Biology, 55.

Thank you for your suggestion. We added a comment addressing this point in lines 167-177.

Sentence 160: Include this reference: Chemical Biology of H2S Signaling through Persulfidation. Milos R. Filipovic, Jasmina Zivanovic, Beatriz Alvarez, and Ruma Banerjee. Chemical Reviews 2018 118 (3), 1253-1337 DOI: n10.1021/acs.chemrev.7b00205

Thank you for your kind suggestion. The reference has been included as well as a brief discussion (lines 183-186).

- Sentence 168-170: Add comment on the absence of the appropriate and reliable method for sensitive and selective determination of H2S and H2S related protein PTMs.

We agree with your comment, thank you for this suggestion. A brief discussion has been added in lines 191-198.

- Sentence 176: Include the references: Vitvitsky V, Miljkovic JL, Bostelaar T, Adhikari B, Yadav PK, Steiger AK, Torregrossa R, Pluth MD, Whiteman M, Banerjee R, Filipovic MR. Cytochrome c Reduction by H2S Potentiates Sulfide Signaling. ACS Chem Biol. 2018 Aug 17;13(8):2300-2307. doi: 10.1021/acschembio.8b00463. Epub 2018 Jul 18. PMID: 29966080; PMCID: PMC6450078.

Domán A, Dóka É, Garai D, Bogdándi V, Balla G, Balla J, Nagy P. Interactions of reactive sulfur species with metalloproteins. Redox Biol. 2023 Apr;60:102617. doi: 10.1016/j.redox.2023.102617. Epub 2023 Jan 27. PMID: 36738685; PMCID: PMC9926313.

As requested, references has been included as well as a comment on the topic (lines 202-215).

- Sentence 189: SQOR is bond to the inner membrane of mitochondria where interacts with CoQ pool that is essential for its activity and acts as a regulator of H2S signaling, discriminating between physiological and poisoning outcomes.

As requested a comment has been added in lines 232-235.

- Sentence 191: Initially, SQOR consumes H2S and oxidised CoQ to transfer the thiol moiety to the reduced GSH thus forming GSSH (glutathione persulfide) and reduced CoQH2.  Alternatively, SQR can use sulfite (SO3) instead of GSH to form thiosulfate (S2O3). Glutathione persulfide is processed via ETHE1 to form sulfite (SO3) which participate in further H2S related catabolic processes in mitochondria.

Consider consulting:

Organization of the Human Mitochondrial Hydrogen Sulfide Oxidation Pathway. Libiad, Marouane et al., Journal of Biological Chemistry, Volume 289, Issue 45, 30901 – 30910.

As suggested, a comment has been added in lines 242-244.

- Sentence 210-215:  Add and comment:

Miljkovic, J. L., Burger, N., Gawel, J. M., Mulvey, J. F., Norman, A. A., Nishimura, T., ... & Murphy, M. P. (2022). Rapid and selective generation of H2S within mitochondria protects against cardiac ischemia-reperfusion injury. Redox Biology, 55.

Thank you for your suggestion. A comment has been added in lines 261-266.

- Sentence: 237: Homocysteinemia-related phenotype should be considered when using the CBS knockout cell model systems?

CBS knockout cell model systems are often used to study the role of CBS in homocysteine metabolism and related pathologies. However, CBS deficiency can also affect other metabolic pathways and lead to changes in cellular metabolism that may not be directly related to homocysteinemia.

- Sentence 251: Hibernation or suspended like animated state?

In response to your request to make it clear whether we are talking about hibernation or an animated state that resembles being suspended, we would like to make it clear that we are referring to a state of suspended animation. We do not want to address the exact physiological mechanisms of hibernation in this section.

- Sentence 457-461: Add references and comments:

Chouchani, E., Methner, C., Nadtochiy, S. et al. Cardioprotection by S-nitrosation of a cysteine switch on mitochondrial complex I. Nat Med 19, 753–759 (2013). https://doi.org/10.1038/nm.3212

Methner C, Chouchani ET, Buonincontri G, Pell VR, Sawiak SJ, Murphy MP, Krieg T. Mitochondria selective S-nitrosation by mitochondria-targeted S-nitrosothiol protects against post-infarct heart failure in mouse hearts. Eur J Heart Fail. 2014 Jul;16(7):712-7. doi: 10.1002/ejhf.100. Epub 2014 May 31. PMID: 24891297; PMCID: PMC4231226.

As requested, a comment has been added in lines 510-518. Thank you for this suggestion.

- Sentence 629-631: Include the references:

Chemical Characterization of the Smallest S-Nitrosothiol, HSNO; Cellular Cross-talk of H2S and S-Nitrosothiols. Milos R. Filipovic, Jan Lj. Miljkovic, Thomas Nauser, Maksim Royzen, Katharina Klos, Tatyana Shubina, Willem H. Koppenol, Stephen J. Lippard, and Ivana Ivanović-Burmazović Journal of the American Chemical Society 2012 134 (29), 12016-12027 DOI: 10.1021/ja3009693

As requested, a comment has been added in lines 681-688.

Eberhardt, M., Dux, M., Namer, B. et al. H2S and NO cooperatively regulate vascular tone by activating a neuroendocrine HNO–TRPA1–CGRP signalling pathway. Nat Commun 5, 4381 (2014). https://doi.org/10.1038/ncomms5381

We added this reference in this section. This same reference has already been discussed and commented in lines 118-127 of the manuscript.

Reviewer 2 Report

To authors:

Major:

This is an interesting and timely review that covers a broad field and the sections on clinical trials are especially noteworthy.  This is the strength of this review.   The authors do a reasonable job in condensing down the background chemistry regarding biosynthesis and metabolism of these compounds, although this is understandably difficult to achieve within the scope of a single article.  The authors are experts in NO biology and, as the authors point out, the CO field is relatively limited, so most of my comments will be addressed to what I believe are several points regarding H2S that need some clarification.

First, biological signaling by H2S is very rare and mainly limited to a few instances where it might reduce disulfide bonds.  Most (essentially all) signaling is from H2S that is oxidized to various forms of persulfides or polysulfides.  This point needs emphasis.

Second, H2S can also be generated by other reactions such as, RSSH + RSH –> RSSR +H2S or from thiosulfate.

Third, The authors point out that most evidence shows that plasma [H2S] is <1 uM, yet they go on to give apparent credibility to a number of studies that report plasma [H2S] in the tens of uM.  These studies are not credible and should not be used in this review.  They promulgate erroneous information and are a disservice to the field.  They should be removed.

Fourth, some of the nomenclature should be revisited.  Although the term’gasotrasmitter’ appears in the literature, it’s use should be discouraged as these molecules do not signal as gasses.  Also, ‘sulfhydration’ should be avoided as there is no hydration reaction, persulfidation is preferred.

Minor:

Line 67: I’m not sure being toxic is a unique requirement for these compounds.  Almost every molecule is toxic at high concentrations.

Line 74: this sounds like a new set of criteria.  What are they?

Fig. 1 shows biosynthesis, but catabolism is as important and should be included along with details on it’s importance.

Line 86: change ‘has been’ to ‘was’

Line 105: By that time...

Line 122: channel’s

Line 166: I believe Whitfield et al (doi: 10.1152/ajpregu.00025.2008) were the first to show that plasma levels over 1 uM were excessive.

Figure 2: There is very little evidence that H2S serves as a ROS scavenger.  Its main effect is to support mitochondrial bioenergetics by feeding electrons into the ETC.

English is fine, only a few minor corrections needed.

Author Response

Reviewer #2

Major:

This is an interesting and timely review that covers a broad field and the sections on clinical trials are especially noteworthy.  This is the strength of this review.   The authors do a reasonable job in condensing down the background chemistry regarding biosynthesis and metabolism of these compounds, although this is understandably difficult to achieve within the scope of a single article.  The authors are experts in NO biology and, as the authors point out, the CO field is relatively limited, so most of my comments will be addressed to what I believe are several points regarding H2S that need some clarification.

Dear referee, we really appreciated the careful revision of our manuscript that will surely benefits by the adjustments you suggested,

You can find a point-by-point list of comments to your requests below.

- First, biological signaling by H2S is very rare and mainly limited to a few instances where it might reduce disulfide bonds.  Most (essentially all) signaling is from H2S that is oxidized to various forms of persulfides or polysulfides.  This point needs emphasis.

Please see the answer to the second request below

- Second, H2S can also be generated by other reactions such as, RSSH + RSH –> RSSR +H2S or from thiosulfate.

To help the reader understand the points and to meet reviewer’s request that were correctly highlighted in the two preceding suggestions, we have added a brief comment from line 183 to 187.

- Third, the authors point out that most evidence shows that plasma [H2S] is <1 uM, yet they go on to give apparent credibility to a number of studies that report plasma [H2S] in the tens of uM.  These studies are not credible and should not be used in this review.  They promulgate erroneous information and are a disservice to the field.  They should be removed.

Thank you for your revision and suggestions. After careful consideration and a thorough discussion amongst all the authors, we have concluded that the field you highlighted is indeed crucial and should be included in the review. We acknowledge that there are still some missing data regarding this evidence, but we believe that the importance of including it outweighs the limitations of incomplete information.

- Fourth, some of the nomenclature should be revisited.  Although the term’gasotrasmitter’ appears in the literature, it’s use should be discouraged as these molecules do not signal as gasses.  Also, ‘sulfhydration’ should be avoided as there is no hydration reaction, persulfidation is preferred. 

We have chosen the term "gasotransmitters" considering the past work of the authors and the fact there are international grants and meeting referring to this group of molecules with this very same word.

Regarding the replacement of sulfhydration with persulfidation, we made the changes accordingly, except when the term “sulfhydration” is specifically reported in the original cited works.

Minor:

- Line 67: I’m not sure being toxic is a unique requirement for these compounds.  Almost every molecule is toxic at high concentrations.

This feature of gaseous signaling molecule has been recognized from the beginning and it’s a characteristic that would rather not avoid.

- Line 74: this sounds like a new set of criteria.  What are they?

We don’t understand the reference probably because of a mistaken line number that could have changed due to formatting procedures. Please refer to the text so that we can identify the issue.

- Fig. 1 shows biosynthesis, but catabolism is as important and should be included along with details on its importance.

We do agree with the importance of catabolism, however the figure is focused on biosynthesis and adding a panel on catabolism’s will make the image less immediate for the reader. Moreover, references in the text about these components are missing for context reasons.

- Line 86: change ‘has been’ to ‘was’

Thank you for the suggestion, the term has been changed accordingly.

- Line 122: channel’s

Thank you for noticing it, the change has been made.

- Figure 2: There is very little evidence that H2S serves as a ROS scavenger.  Its main effect is to support mitochondrial bioenergetics by feeding electrons into the ETC.

We strongly agree with you, thus we entitled the figure “main targets of the three gasotransmitters in cardiovascular scenario”. However, we added a sentence in the text to highlight this point.
